# Mechanisms of Bioactive Glass on Caries Management: A Review

**DOI:** 10.3390/ma12244183

**Published:** 2019-12-12

**Authors:** Lin Lu Dai, May Lei Mei, Chun Hung Chu, Edward Chin Man Lo

**Affiliations:** 1Faculty of Dentistry, University of Hong Kong, Pok Fu Lam, Hong Kong; doreen07@hku.hk (L.L.D.); chchu@hku.hk (C.H.C.); 2Faculty of Dentistry, University of Otago, 9054 Dunedin, New Zealand; may.mei@otago.ac.nz

**Keywords:** bioactive glass, mechanism, caries, review

## Abstract

This review investigates the mechanisms of bioactive glass on the management of dental caries. Four databases (PubMed, Web of Science, EMBASE (via Ovid), Medline (via Ovid)) were systematically searched using broad keywords and terms to identify the literature pertaining to the management of dental caries using “bioactive glass”. Titles and abstracts were scrutinized to determine the need for full-text screening. Data were extracted from the included articles regarding the mechanisms of bioactive glass on dental caries management, including the aspect of remineralizing effect on enamel and dentine caries, and antimicrobial effect on cariogenic bacteria. After removal of duplicates, 1992 articles were identified for screening of the titles and abstracts. The full texts of 49 publications were scrutinized and 23 were finally included in this review. Four articles focused on the antimicrobial effect of bioactive glass. Twelve papers discussed the effect of bioactive glass on demineralized enamel, while 9 articles investigated the effect of bioactive glass on demineralized dentine. In conclusion, bioactive glass can remineralize caries and form apatite on the surface of enamel and dentine. In addition, bioactive glass has an antibacterial effect on cariogenic bacteria of which may help to prevent and arrest dental caries.

## 1. Introduction

Dental caries is a prevalent oral disease worldwide and can occur in both primary and permanent dentitions throughout an individual’s life. It is a biofilm-mediated disease, resulting in mineral loss and destruction of dental hard tissues. Cariogenic bacteria will produce acids, causing prolonged periods of low pH environment in the oral cavity which leads to demineralization of dental hard tissues [1]. The process of dental caries starts with chemical dissolution of enamel and dentine caused by the acids produced by the bacteria that adhere onto the tooth surface. If sufficient time is allowed for progression, the carious lesions on the surface will progress to cavity formation in the affected tooth [2,3]. 

The current approaches of caries management aim to 1) stop or control the progression of caries, 2) preserve dental hard tissue as much as possible, and 3) avoid the re-restoration process [4]. Management of carious lesions with varying severity is outlined below. For initial lesions, nonsurgical approaches are commonly used. Fluoride-containing products are delivered in different forms onto teeth to promote remineralization and the mineral contents of the lesions are recovered by penetration of calcium and phosphate from a higher concentration into the lesions. Casein phosphopeptide–amorphous calcium phosphate (CPP-ACP) is a stabilized system of Ca–P that has superior remineralization potential on carious lesions. A modification is to add fluoride into the system (CPP-ACPF), which can improve the remineralization efficacy compared to that of the original system [5]. The approaches listed above are mainly due to their remineralization effect on the tooth surface. Some anticaries agents possess antibacterial properties, which can inhibit the growth of cariogenic bacteria. Chlorhexidine (CHX) is a type of antibacterial agent which can reduce the *Streptococcus mutans* level in the oral cavity [6]. Triclosan, which can affect the acid production of biofilm, is another anticaries agent. Previous studies have shown that amino acid arginine has an anticaries effect because of its effect on oral biofilms. Furthermore, xylitol is a natural substitute for sugar and has antibacterial potential on dental caries. Similarly to the other two previously mentioned agents, they possess the ability to control bacterial level, thus promoting the process of remineralization [3,5].

Placement of pit and fissure sealant is another minimally invasive therapy for managing initial lesions of the tooth surface. For moderate lesions, mechanical blocking or sealing off the lesion is an effective method to arrest caries after applying resin-based fissure sealant. Topical application of silver diamine fluoride (SDF) is an alternative way to arrest moderate carious lesions due to its antibacterial effect and remineralization effect [2,3]. Besides, the classic standard treatment of extensive lesions is removing all the demineralized tissues of the tooth and placing dental restorative material like composite resin to fill up the prepared cavity. In a recent development, stepwise or partial removal of caries is a new trend to preserve more dental tissues and it can reduce the incidence of pulpal exposure and favor the formation of tertiary dentine after restoration [2,3]. For the restorative method mentioned above, various materials are used. These include chemically bonded ceramic cements set by acid–base reaction, such as zinc phosphate, silicate, polycarboxylate, and glass ionomers. Composite resin is another type of cement that is set by a polymerization reaction. In addition, resin-modified glass ionomer cement is made by combining the two reactions [7]. 

Bioactive glass is a relatively new agent with an ability to heal bone defects caused by trauma or diseases and lead to bone regeneration. It has been applied in many healthcare fields. The first bioactive glass introduced in 1969 was a sodium, calcium, and phosphorus silicate glass. Currently, there are different types of bioactive glass, such as silicate-based glass and phosphate-based glass. Bioactive glass is an excellent material from the perspective of material properties. Because of its bioactivity and biocompatibility, the basic concept of applying bioactive glass in bone repair is to use a scaffold to act as a 3-dimensional template to guide bone regeneration [8]. It has been applied in wide-ranging fields, especially in the use of bone grafts, scaffold, disinfectant of the dental root canal and coating materials of dental implants [9]. The main advantage of bioactive glass in bone augmentation and repair is its high reactivity when in contact with bone surface and the most well-known capability of bioactive glass is the bonding ability to bone as well as stimulation of bone growth [10]. Firstly, when the material is in contact with an aqueous solution, the particles will change to mesoporous shape. Then, the particles will form an enrichment layer to produce an apatite-like layer on bone surface, similar to the component of bone or other hard tissues [11]. The formation of a hydroxyapatite (HA) layer involves the exchange of ions between the bioactive glass and the bone surface. The deposition of bone-like precipitates on bone surface plays a key role in the healing of bone defects [9]. The action on tooth is similar to that on bone. Bioactive glass can mineralize dentine tubules to relieve tooth sensitivity. The process is as follows: The glass material dissolves into an aqueous solution, followed by a pH rise. The pH rise promotes precipitation of hydroxyapatite (HA), the main component of mineralized enamel and dentine. Calcium and phosphate ions from bioactive glass and mineralizing agents in saliva may enhance the process of mineralization [8]. The most successful commercial product derived from a type of noncrystalline amorphous bioactive glass (Bioglass 45S5) with the name of NovaMin (GlaxoSmithKline, UK) is used in dentine repairing toothpaste, which can relieve the symptoms of dentine hypersensitivity. Bioglass 45S5 is silica-based and composed of 45 wt% SiO_2_, 24.5 wt% CaO, 24.5 wt% Na_2_O, and 6.0 wt% P_2_O_5_. It can appear in the form of particulates or granules [12,13,14]. 

Although studies have shown that bioactive glass has an ability to promote regeneration of bone and mineralization of dental hard tissues, it is not known whether bioactive glass is effective in preventing and arresting dental caries. Literature reviews conducted so far focus mainly on the mechanisms of bioactive glass on bone regeneration, tissue engineering, or dentine hypersensitivity, and very few have reviewed the mechanisms of action of bioactive glass on caries management. The purpose of this study was to review the literature on the actions of bioactive glass on dental caries management regarding its effects on the caries process and cariogenic bacteria. 

## 2. Materials and Methods

### 2.1. Searching Strategy

The literature search was conducted on four databases, namely PubMed, Web of Science, EMBASE (via Ovid), and Medline (via Ovid). Articles in these databases were searched using the keywords (“bioglass” OR “bioactive glass” OR “bioceramic”) AND (“dentistry” OR “dental caries”). 

### 2.2. Study Inclusion and Exclusion

The lists of publications from the four databases were checked to remove duplications. Afterwards, the titles and abstracts of the identified articles were screened. This review aimed to summarize the mechanisms of bioactive glass on caries management. The inclusion and exclusion criteria were as follows.

#### 2.2.1. Inclusion Criteria: 

Laboratory studiesStudies related to the antimicrobial effect of bioactive glassStudies on the remineralization effect of bioactive glass on dental hard tissues (enamel and dentine)

#### 2.2.2. Exclusion Criteria: 

Studies on root canal therapy and pulp regenerationStudies on periodontal diseaseStudies on orthodontic treatmentStudies on tissue engineeringStudies on bioactive composites or other bioactive materials

Two reviewers independently performed the screening to select potentially relevant articles. An independent reviewer was consulted on studies that were not able to be determined. The information extracted after reading the full text of the selected articles included basic publication details (authors and year), methods and materials used, measurement of outcomes, and main results.

## 3. Results

A total of 1992 potentially eligible articles published up to July 2019 (1051 articles in PubMed, 437 in Medline, 253 in Web of Science, and 251 in Embase) were identified (Figure 1). After checking for duplications, 748 records were removed. For the remaining 1244 articles, titles and abstracts were screened and they were classified into randomized clinical trial (RCT), case report, literature review, and laboratory study. Only laboratory studies were selected, and studies not related to the mechanisms of bioactive glass on caries management were excluded. Full-text readings were carried out on 49 articles and only 23 articles met the study eligibility criteria to be included in the final review. Among these 23 publications, there were 4 studies which examined the action of bioactive glass on cariogenic bacteria (Table 1), 12 studies focused on the remineralizing effect of bioactive glass on enamel (Table 2), while 9 studies investigated the effect of bioactive glass on dentine mineral contents (Table 3).

### 3.1. Effect of Bioactive Glass on Cariogenic Bacteria

Table 1 shows the main findings of the four studies that investigated the effect of bioactive glass on cariogenic bacteria. A study found that the minimal inhibitory concentration (MIC) and minimal bactericidal concentration (MBC) of bioactive glass powder (45S5; Datsing Bio-Tech Co. Ltd, Beijing, China) were 18.8 and 37.5 mg/mL, respectively. The study showed that when bioactive glass dissolved in water, alkaline ions were released to raise the pH of the solution and this could kill *Streptococcus mutans* [15]. Another type of bioactive glass-ceramic (Biosilicate) was shown to exhibit antimicrobial properties that could inhibit a wide spectrum of microorganism. It was found that Biosilicate possessed antibacterial action against multi-species cariogenic bacteria strains, such as *Streptococcus mutans*, *Actinomyces naeslundii*, *Lactobacillus casei*, through agar diffusion and direct contact [16]. Another investigation found that growth of *L. casei* incubated in a silver-doped bioactive glass (Ag-BGN@MSN) was inhibited. Silver melted into bioactive glass displays a synergistic effect on microorganisms as it can inhibit the growth of cariogenic bacteria [17]. A study investigated cation-doped (Ag, Mg, Sr, Zn, and Ga) bioactive ceramics and revealed a bacterial inhibitory effect on *Streptococcus mutans* and *Lactobacillus casei* with varied MIC [18].

### 3.2. Effect of Bioactive Glass on the Mineral Content of Enamel and Dentine

Table 2 shows the main findings of the 12 published papers that investigated the effect of bioactive glass on the mineral content of enamel. Demineralized enamel was treated with different types of bioactive glass. Surface microhardness of the enamel tissue decreased after the demineralization procedure and increased after application of bioactive glass. The value of microhardness was found to be higher in bioactive glass group when compared to the control group or application of other agents [19,20,21,23,29,38]. A study reported the recovery rate of microhardness on demineralized enamel surface after treatment with bioactive glass was 28.8% [21]. Further investigation showed that a combination of bioglass paste and cold plasma had a synergistic effect on increasing the surface microhardness of demineralized enamel [29]. Apart from assessing microhardness, the mean carious lesion depth in specimens treated with bioactive glass were significantly lower than those of specimens without bioactive glass treatment [20,26,38]. A study assessed the percentage of regain in lesion depth after remineralization and the experimental group with bioactive glass had the highest regain percentage (73.0 ± 3.0%) of lesion depth in enamel [22]. In addition, energy-dispersive X-ray spectroscopy (EDX) analysis indicated that Ca/P ratio was higher in the region treated with bioactive glass than in other regions not covered by bioactive glass particles [26,29,30]. One study found that compared to just application of deionized water, enamel lesions after treated with two bioactive glass had significantly higher Ca/P ratios [30].

Table 3 shows the main findings of the nine studies on the effect of bioactive glass on the mineral content of dentine. Microhardness measurement was a commonly used method to evaluate the surface of the demineralized dentine. The remineralization process induced an increase in surface microhardness of carious lesions [32]. In addition, it was found that application of Bioglass 45S5 significantly increased root dentine microhardness [36]. Dentine lesion depth decreased after the application of bioactive glass in two in vitro studies [17,20]. Another study used visual–tactile examination to assess the severity of root caries and found that there was a significant reduction in the group combining bioglass and fluoride and that group also had the highest percentage (60%) increase in mineral deposition [31]. Dentine discs treated with bioactive glass had significantly higher mineral matrix area ratio when compared to that of discs in the artificial saliva and DL-aspartic amino groups [37]. Furthermore, weight loss of dentine slices treated with BAG S53P4, an amorphous glass with the composition of 53 wt% SiO_2_, 23 wt% Na_2_O, 20 wt% CaO, and 4 wt% P_2_O_5_, was less than that of slices without such treatment [34]. EDX was used in a study to analyze the elements in the occluding materials within dentine tubules. The results indicated that the ratio of Ca/P of hydroxyapatite was not significantly different between the bioactive glass and control groups [39]. 

Apart from the approaches mentioned above, qualitative parameters were also used to measure the mineralization effect of bioactive glass on enamel and dentine. Most of these studies [25,26,27,29,33,35,37] analyzed the morphology of enamel and dentine surface using scanning electron microscopy (SEM). The deposits newly formed on the surface of dental hard tissue were crystal-like hydroxyapatite (HAP) and rich in calcium and phosphate with the presence of silica. A layer of mineral formed by the particles of bioactive glass covered the lesion surface in the remineralized enamel group [24]. Different from that seen on enamel, a layer of particles of bioactive glass not only deposited on the dentine surface, but also partially or completely occluded dentine tubules during remineralization [33,39]. Figure 2 shows two SEM images of demineralized dentine with or without treatment with bioactive glass. After observing the morphology of the remineralized enamel and dentine, the content of the new deposition was assessed by X-ray diffraction (XRD) [25]. XRD results showed that the bioactive glass (Ag-BGN@MSN) particle used had an amorphous two-dimensional hexagonal structure [17]. Another XRD study found that both nanoparticles and conventional bioactive glass were in amorphous state [35]. Furthermore, XRD results of another two studies matched the standard diffraction peak of hydroxyapatite crystal on the enamel and dentine surfaces [31,37]. Strontium-modified bioactive glass displayed a higher intensity of XRD peaks than that of the original bioactive glass [33]. Another study used a qualitative method to assess the mineral concentration by using X-ray microtomography. The result showed that the highest percentage increase of mineral content in the lesion area that was treated with bioglass and fluoride [31]. Raman spectroscopy was another method used in the studies to confirm the content of remineralized enamel and dentine tissues. The phosphate peak of sound enamel and dentine appeared in a specific wavelength (around 960 cm^−1^) of Raman spectra, while demineralized hard tissues showed no peaks. In two studies, the dental tissues treated with bioactive glass displayed the intensity of phosphate peak [26,32], while another study illustrated that there was a reduction of the intensity of phosphate peak in demineralized enamel compared to sound enamel [24]. Demineralized dentine showed phosphate peak after one-day treatment with nanoparticle bioactive glass, but no phosphate peak appeared after treatment with conventional bioglass, though all the dentine specimens immersed in the two types of bioactive glass had deposition of apatite on the surface after 10 or 30 days [35]. 

### 3.3. Effect of Bioactive Glass on the Organic Content of Dentine 

Only two studies mentioned changes in the organic content of dentine. One of them found that the dentine remineralized in nanometric bioactive glass suspension, compared to the dentine in PeioGlas^®^ (Millipore, Bedford, MA, USA), Bioglass 45S5 with particles size ranging from 90 to 710 μm showed a significantly lower protein content due to the removal of organic contents [35]. As illustrated in another study, Raman spectra showed no peak for hydroxyapatite but only a high intensity of organic components in the demineralized dentine without treatment by the bioactive glass [32]. Lower organic content indicates better remineralization.

## 4. Discussion

After screening and analyzing the results of all selected laboratory studies, a number of possible mechanisms of how bioactive glass act on dental caries were found. The mode of action of bioactive glass for arresting caries is related to two aspects: 1) the antibacterial properties of bioactive glass on cariogenic bacteria, and 2) the remineralizing effect on the mineral content of dental hard tissues.

In the oral cavity, oral microbiota and dental biofilms are commonly present. Formation of dental plaque (dental biofilm) involves several stages. First, the acquired pellicle on tooth surface provides sites for bacterial colonizers. The oral microorganisms then grow and form a conditioning film of bacteria, proteins, and other bacterial products covering the tooth surface. *Streptococci*, *Lactobacilli,* and *Actinomycetes* are recognized as the main species of bacteria contributing to caries progression. *Streptococci* have high incidence and proportions and in the dental biofilms covering early caries lesions [2]. The key microorganism in initiating and developing dental caries is *Streptococcus mutans* (*S. mutans*). *Lactobacillus casei* (*L. casei*) is a type of cariogenic bacteria strains that commonly appear in deep or advanced caries lesions. More recently, another type of acid-producing and acid-tolerating species, called *Actinomycetes*, has been found to be associated with caries [40]. This systematic review found that very few studies investigated the antimicrobial effect of bioactive glass. This may be because the most obvious advantage of bioactive glass is its remineralization effect on bone and teeth rather than its bactericidal efficacy. Xu et al. assayed plaque biofilm of *S. mutans* and applied bioactive glass 45S5 at a concentration twice of the minimal bactericidal concentration to show that the bioactive glass had a great inhibitory effect on *S. mutans* biofilm [15]. This suggests that the concentration of antimicrobial agent needed for inhibiting biofilm may be many times higher than that for inhibiting planktonic bacteria. The possible action of bioactive glass acting on cariogenic bacteria is release of alkaline ions, followed by pH elevation that builds an environment in which bacteria cannot grow. This is similar to the mechanism of action of arginine, an amino acid, in which the arginine deiminase system has been identified as a novel technology to prevent initiation of the dental caries process by increasing pH around the biofilm on tooth surface [3]. Apart from the process of pH elevation, the presence of antibacterial ions can also control bacterial growth. Cation-doped bioactive ceramics, such as Ag, Mg, Sr, and Zn, have good inhibitory effect on *S. mutans* and *L. casei* [18]. Two literature reviews proposed that silver diamine fluoride (SDF), in which silver ion is the major antimicrobial agent, is an effective treatment to arrest established dental caries [3,40]. It has been shown by utilizing bacterial and biofilm models, that SDF can inhibit the growth both *Streptococcus mutans* and *Actinomyces naeslundii* [41]. Therefore, bioactive glass with silver may have additional inhibition effect against cariogenic bacteria. 

The various compositions in bioactive glass have different roles in the remineralization process. The proportion of calcium and phosphate in dental tissues is identical to that in bone. Phosphate has a great contribution to hydroxyapatite formation and increases biocompatibility significantly. Formation of hydroxyapatite layer promotes remineralization in enamel and dentine. The physical occlusion on the lesion surface begins with the bioactive glass particles exposed to the aqueous environment, along with ion release and pH elevation [37]. When the biomaterial is exposed to an aqueous environment, sodium ions will exchange with H^+^ (hydrogen ions). Meanwhile, Ca^2+^ (calcium ions) in the particles as well as PO_4_^3−^ (phosphate ions) are released from the biomaterial. Thus, a localized pH rise will allow the precipitates of calcium and phosphate ions, together with the ions from saliva to form a calcium phosphate (Ca–P) layer on the lesion surface [20]. The silica network from bioactive glass can react with hydroxyl ions from aqueous solution and form soluble silanol compounds. It can be observed that the increase in Ca and P content would induce a decrease in Si content [29]. The newly formed layer displays good resistance to abrasion and transforms to a hydroxyapatite layer ultimately, which is structurally similar to those of original enamel and dentine [32]. 

Topical fluoride has already been proved to be effective in treating dental caries. The mechanism of fluoride is to inhibit demineralization and promote remineralization, which conducts a similar procedure with bioactive glass. The fluoride in oral fluid or solution can penetrate along with the acid at the subsurface and protect the minerals from dissolution, and thus prevents the demineralization process. After acidic challenge, fluoride will be adsorbed to the demineralized crystals and attract calcium ions, thus making the solution highly supersaturated with respect to fluorohydroxyapatite, which can promote the remineralization process [42]. A recent review found that SDF can inhibit the demineralization and promote remineralization of the mineral content of enamel and dentine and protect collagen matrix from degradation [40]. An in vitro study showed that the fluoride in bioactive glass could be switched to fluorapatite on the tooth surface, which leads to higher resistance to acid dissolution [20]. The precipitation of mineral deposits occurs mostly in the superficial layer, particularly when fluoride is present [31]. The deposition of a fluoride-contained mineral layer on dentine surface can occlude dentine tubules and reduce permeability [21].

A study stated that strontium can be a substitution of calcium in bioactive glass which may show a better bonding ability [9]. Strontium can supply ions for hydroxyapatite formation. Incorporation of strontium and fluoride can inhibit hydroxyapatite dissolution by the acids produced by cariogenic bacteria. Strontium can be a substitute for calcium for precipitate formation and it has synergistic caries inhibition effect with fluoride. The remineralization effect can last for different periods due to the addition of various proportions of strontium into the bioactive glass, which shows that strontium may be a beneficial factor in preventing caries through remineralizing dental hard tissues [33]. Besides, nanometric particles of bioactive glass have better remineralization potential compared to the conventional ones because of its larger surface area and higher Ca/P ratio [22]. The experiments conducted by Meret showed a greater effect of remineralization on dentine surface due to the nanosize of bioactive glass [35], while another study also demonstrated that Biosilicate microparticles were more effective in slowing down progression of caries lesions and promoting remineralization [38]. Smaller particles may completely block the porosity of enamel and dentine lesions. These microstructures are capable of penetrating from the tooth surface to the whole lesion and enhancing the remineralization of carious lesions [24].

An advantageous aspect of bioactive glass is its bioactivity and biocompatibility. Previous studies adopted the direct contact cell viability method to evaluate the biocompatibility of bioactive glass and showed a high cell survival rate [43,44]. As a very safe material and based on the merits stated above, a potential new application of bioactive glass is for dental caries prevention and remineralization of early caries lesions [45]. Further research should pay more attention to how the bioactive glass work in treating dental caries in the real oral environment.

## 5. Conclusions

Based on the findings of the present review, it is concluded that bioactive glass is able to inhibit the growth of cariogenic bacteria. Bioactive glass can promote remineralization by forming apatite on the surface of demineralized enamel and dentine. The main mechanisms of bioactive glass for caries management include an antibacterial effect on cariogenic bacteria, prohibition of mineral demineralization, and promotion of remineralization. 

## Figures and Tables

**Figure 1 materials-12-04183-f001:**
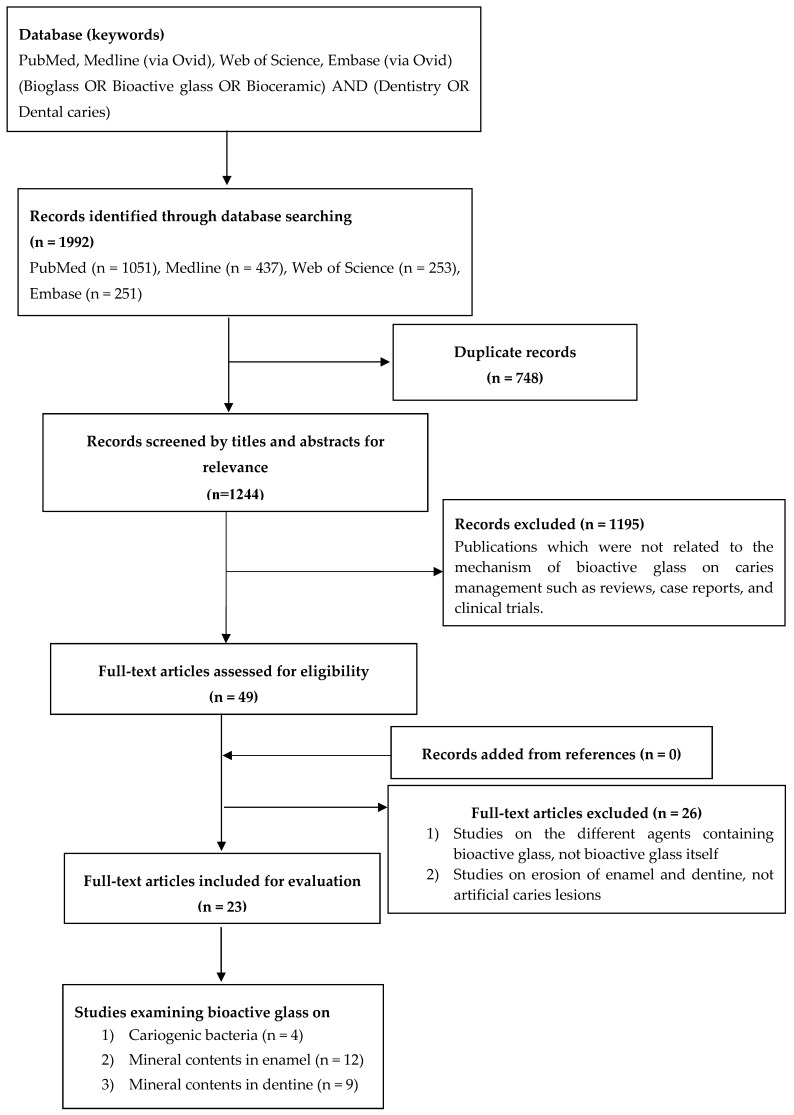
Flowchart of literature search of bioactive glass.

**Figure 2 materials-12-04183-f002:**
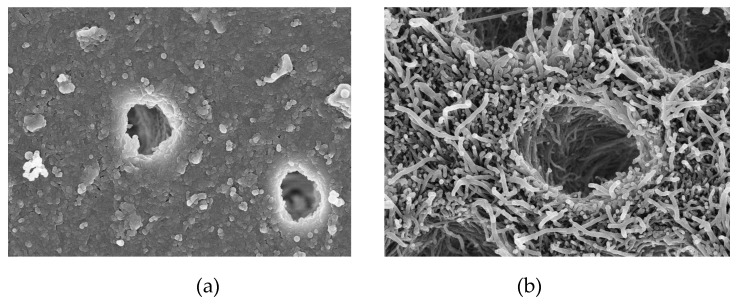
SEM images of the morphology of demineralized dentine: (**a**) 10,000× magnification image of demineralized dentine treated with bioactive glass; (**b**) 10,000× magnification image of demineralized dentine without treatment with bioactive glass.

**Table 1 materials-12-04183-t001:** Summary of studies on antimicrobial effect of bioactive glass.

Author (Year)	Methods	Main Findings
Xu et al. (2015) [15]	MIC and MBC were determined to test the antibacterial effect of a bioactive glass against *Streptococcus mutans*.	The MBC and MIC of bioactive glass was 37.5 and 18.75 mg/mL, respectively.
Martins et al. (2011) [16]	Three methods (agar diffusion, direct contact, and MIC) were used to determine the antibacterial effect of a bioactive glass-ceramic (Biosilicate) against a wide spectrum of bacteria. The assessed cariogenic species were *Streptococcus mutans*, *Lactobacillus casei*, *Actinomyces naeslundii*).	The MIC of Biosilicate ranged from ≤ 2.5 mg/mL to 20 mg/mL in different bacterial species. The best antibacterial effect of Biosilicate was against *S. mutans* (inhibition halo: 19.0 ± 2.0 mm) and *S. mutans* clinical isolate (MIC ≤ 2.5 mg/mL).
Jung et al. (2018) [17]	Light absorbance was used to evaluate the antibacterial effect of silver-doped bioglass MSN against *Lactobacillus casei*.	The increasing density of silver-doped bioglass MSN induced reduction of light absorbance. It illustrated that bacterial growth was inhibited.
Siqueira et al. (2019) [18]	Agar dilution method was used to determine the MIC values. The assessed cariogenic species were *Streptococcus mutans* and *Lactobacillus casei*.	Both the MIC of Bioglass and Biosilicate against *S. mutans* were 4mg/mL, which was the same as the MIC against *L. casei*. Bio-FP doped with different cations had different MIC against *S. mutans* and *L. casei*: Ag (8 and 4 mg/mL), Mg (2 and 4 mg/mL), Sr (2 and 4 mg/mL), Zn (2 and 4 mg/mL), Ga (2 and 4 mg/mL).

CFU, colony-forming units; MIC: minimal inhibitory concentration; MBC: minimal bactericidal concentration; TCS, triclosan; MSN, mesoporous silica nanoparticle.

**Table 2 materials-12-04183-t002:** Summary of effect of bioactive glass on enamel mineral content.

Author (Year)	Methods	Main Findings
Palaniswamy et al. (2015) [19]	Demineralized enamel was treated with ACP-CPP and BAG, followed by microhardness test. BAG and ACP-CPP were applied on samples for 10 days in the first remineralization cycle and applied for another 5 days in the second remineralizing cycle.	Microhardness of dentine treated with ACP-CPP and BAG both increased but showed no significant difference between the 1st and 2nd remineralization cycles (BAG after 10 days: 346 ± 45; BAG after 15 days: 363 ± 65).
Rajan et al. (2015) [20]	Demineralized teeth were allocated into five groups as follows: fluoridated toothpaste, CPP-ACPF, ReminPro, SHY-NM and control group. Micro-CT was used to measure lesion depth.	Lesion depth after remineralization in SHY-NM group showed the least mean score of 987 µm compared to other groups.
Soares et al. (2017) [21]	Enamel samples with artificial lesions were treated with CPP-ACP, BAG, ReminPro, and self-assembling peptide. The recovery rate of microhardness was assessed.	Microhardness recovery rate of enamel treated with peptide was the highest (62.1%), followed by CPP-ACPF (48.4%) and BAG group (28.8%).
Prabhakar et al. (2009) [22]	Teeth with artificial carious lesions were divided into 2 experimental groups (sodium fluoride films, bioactive glass films) and 2 control groups (control films placed interproximally and no treatment group).	Percentages of regain of lesion depth after remineralization in BAG were more in the experimental groups (NaF films: 67.7% ± 3.8%; and BAG films: 73.0% ± 3.0%) than those in the control groups (control film: 21.1% ± 3.3%; and no treatment: 30.7% ± 2.5%).
Chinelatti et al. (2017) [23]	Artificial caries lesions were formed on enamel fragments and either treated with Biosilicate or acidulated phosphate fluoride (APF), or had no treatment (control), followed by microhardness test.	Biosilicate group had higher microhardness on enamel surface (265 ± 10 KHN) than APF and control group. CLSM also displayed shallower lesions in Biosilicate group when compared to APF and control group.
Milly et al. (2013) [24]	Enamel samples with artificial WSLs were assigned to 4 groups: BAG slurry, PAA-BAG slurry, remin solution, and deionized water; the surface and cross-sectional microhardness of enamel was assessed.	BAG group illustrated the highest surface microhardness (138 ± 5 KHN), but there were no significant differences among the other groups.
Bakry et al. (2014) [25]	Demineralized enamel specimens were divided into 4 groups: (1) no intervention, (2) only bioglass, (3) only brushing abrasion challenge, and (4) bioglass + brushing abrasion. After demineralizing and application of bioglass, all specimens were stored in remineralizing medium for 24 h, followed by removing the thin layer of bonding agent on bioglass in Groups 2 and 4, and then Groups 3 and 4 were sent to brushing abrasion challenge.	Hydroxyapatite was detected using XRD on the surface of enamel in Group 2 and Group 4 and these two groups also exhibited 100% coverage of crystalline structures on enamel surface.
Zhang et al. (2018) [26]	Artificial enamel WSLs were assigned to BG slurry, BG+PAA, CS-BG, CS-BG+PAA, remin solution, and deionized water groups. Microhardness was assessed and the intensity of surface mineral content was measured by Raman intensity mapping.	Intensity increase in BG group was significantly greater when compared to those without BG. CS-BG+PAA group showed the highest microhardness (222 ± 38 KHN) of enamel surface. Other groups with BG also exhibited higher microhardness than the control group.
Narayana et al. (2014) [27]	Enamels with artificial carious lesions were treated with bioactive glass, fluoride toothpaste, CPP-ACP, or CPP-ACPF and the control had no treatment. EDS was used to test the weight change of different elements.	BAG group showed significant difference when compared with control group for elements Ca and P. The mean weight percentage of Ca was 40.0% (BAG) and 31.1% (control), while the percentage of P was 14.0% (BAG) and 13.2% (control).
Mehta et al. (2014) [28]	Enamel specimens were randomly distributed into two groups: BAG and CPP-ACP dentifrice. Vickers microhardness test was used.	Mean microhardness values were 372 VHN in BAG group and 357 VHN in CPP-ACP group afterremineralization, but the difference was not significant.
EI-Wassefy et al. (2016) [29]	Demineralized enamels were treated with no treatment, fluoride varnish, cold plasma, bioglass paste, cold plasma + bioglass paste. Microhardness was assessed by Vickers hardness tester.	Microhardness of enamel surface become higher in PB groups (175 VHN and 221 VHN) when compared with bioglass groups (153 VHN and 201 VHN) at 30 and 50 µm depth, but with no significant difference between the two groups at 70–200 µm depth.
Zhang et al. (2019) [30]	Enamel slabs with artificial WSL were assigned into 4 groups: bioglass (chitosan pre-treated lesions), chitosan-bioglass slurry, remin solution (PC), and deionized water (NC). Subsurface microhardness was assessed.	Mean hardness of bioglass group and chitosan–bioglass group were 56.7 ± 8.7 and 65.1 ± 8.9 KHN, which were significantly higher than those of NC group (12.7 ± 1.3 KHN) and PC group (18.6 ± 5.8 KHN).

CPP-ACP, calcium phosphate–casein phosphopeptide; SHY-NM: name of a bioactive glass; HA, hydroxyapatite; PAA-BAG, bioactive glass containing polyacrylic acid; WSL, white spot lesions; EDS: energy dispersive X-ray spectroscopy; CLSM, confocal laser scanning microscopy analysis; APF, acidulated phosphate fluoride; XRD, X-ray diffraction.

**Table 3 materials-12-04183-t003:** Summary of effect of bioactive glass on dentine mineral content.

Author (Year)	Methods	Main Findings
Sleibi et al. (2018) [31]	Teeth with root caries were divided into 4 groups and treated with different agents (CPP-ACP+fluoride, bioglass+fluoride, fluoride only, no treatment). Severity index of root caries was evaluated through visual–tactile examinations. X-ray microtomography was used to measure mineral change.	The bioglass and fluoride group had the maximum reduction (100%) in severity index of root caries and it also had the highest percentage (60%) increase in mineral deposition.
Rajan et al. (2015) [20]	Demineralized teeth were treated with fluoridated toothpaste, CPP-ACPF, ReminPro, SHY-NM (bioglass), and no treatment (negative control). Lesion depth was measured after application.	SHY-NM (bioglass) group showed the lowest mean lesion depth after remineralization procedure.
Sauro et al. (2011) [32]	Dentine segments were treated with bioactive glass (Sylc), NaH C_2_O_4_ H_2_O, Cavitron Prophy Powder, EMS Perio, CPP-ACP, Colgate Sensitive Pro-Relief, NUPRO Solution Prophy Paste. Microhardness and EDX were evaluated.	The dentine surface hardness increased after treated with bioactive glass (Sylc). There was no significant change in Ca and P/O ratios.
Saffarpour et al. (2017) [33]	Demineralized dentine discs were treated with 3 agents: bioactive glass (BG), BG modified with 5% strontium, BG modified with 10% strontium and followed by evaluation of morphology.	BG with 10% strontium showed highest rate of remineralization and completely occluded dentinal tubules.
Forsback et al. (2004) [34]	Dentine discs were treated with bioactive glass S53P4 and control glass (CG). Weight loss of dentine discs was measured by weighing before and after remineralization.	Weight loss was less when discs were pretreated with BAG (21.0 ± 7.4 µg/mm^2^) than without BAG (49.1 ± 6.5 µg/mm^2^).
Vollenweider et al. (2007) [35]	Demineralizing dentine bars were applied by nanometric bioactive glass (NBG) and PeriGlas (PG) suspension. SEM was used to observe the dentine surface.	Dentine specimens treated with NBG showed apatite depositions on the surface after 10 or 30 days.
Jung et al. (2018) [17]	Demineralized dentine discs were divided into four groups: bioglass, MSN, silver-doped bioglass MSN, and no treatment, followed by acid resistance test.	Silver-doped bioglass MSN group had dentinal tubules completely occluded to a depth of 2–3 µm and the highest proportion (83.4% ± 7.5%) of occluded area after acid challenge.
Cardoso et al. (2018) [36]	Root dentine slices were allocated into four groups: MTA, ERRM, Bioglass 45S5, and NbG. Microhardness was assessed.	Bioglass 45S5 group showed an increase in microhardness.
Zhang et al. (2019) [37]	Dentine discs treated with EDTA were allocated to 4 groups: AS (artificial saliva), Asp, BAG, Asp-BAG, and followed by 6% citric acid challenge. The mineral matrix ratio was measured.	Compared to AS and Asp group, BAG group (17.8 ± 2.3) and Asp-BAG group (12.5 ± 2.3) had significantly higher mineral matrix area ratio.

EDX, energy dispersive X-ray spectroscopy; Ag-BGN, silver-doped bioactive glass; MSN, mesoporous silica nanoparticle; DW: deionized water; EDTA, ethylene diamine tetraacetic acid; Asp, DL-aspartic amino; MTA, mineral trioxide aggregate; ERRM, EndoSequence Root Repair Material; NbG, niobophosphate glass.

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
