# Peer review of "Mechanisms of Bioactive Glass on Caries Management: A Review"

_materials, 2019, doi:10.3390/ma12244183_

Round 1

Reviewer 1 Report

Please add more explanation about Bioglass 45S5 (page 2,line 81). For example, what are the compositions? What was the size or shape? Are they granules? What is the BAG S53P4? (page 11,line 230) What is PeioGlass? (page 12, line 264)

Author Response

Points:

Please add more explanation about Bioglass 45S5 (page 2,line 81). For example, what are the compositions? What was the size or shape? Are they granules? What is the BAG S53P4? (page 11,line 230) What is PeioGlass? (page 12, line 264)

Response:

Thank you for your comments and suggestions.

As suggested, more explanation on the bioactive glasses is given:

- In page 2, line 87-90, “The most successful commercial product derived from a type of non-crystalline amorphous bioactive glass (Bioglass 45S5,) with the name of NovaMin (GlaxoSmithKline, UK) is used in dentine repairing toothpaste, which can relieve the symptom of dentine hypersensitivity. Bioglass 45S5 is silica-based and composed of 45 wt% SiO2, 24.5 wt% CaO, 24.5 wt% Na2O, and 6.0 wt% P2O5. It can appear in the form of particulates or granules.” One more reference (now numbered 14) is added.

- In page 11, line 235-236, added “BAG S53P4, an amorphous glass with the composition of 53 wt% SiO2, 23 wt% Na2O, 20 wt% CaO and 4 wt% P2O5,”

- In page 12, line 274-275, added “Bioglass 45S5 with particles size ranging from 90 to 710 μm.”

A revised manuscript is attached.

Reviewer 2 Report

Manuscript title: Mechanisms of bioactive glass on caries management: a review

Dear Authors,

Thank you for this submission. This article is interesting and contains useful information.  There is a continuum interest in assessing and evaluating new or existing materials which may help to control oral diseases.

The use of bioactive materials is well reported. This review focuses on the whether bioactive glass is effective in preventing and arresting dental caries. Literature reviews conducted so far focus mainly on the mechanisms of bioactive glass on bone regeneration, tissue engineering or dentine hypersensitivity and very few reviewed the mechanisms of actions of bioactive glass on caries management. The purpose of this study was to review the literature on the actions of bioactive glass on dental caries management regarding its effects on the caries process and cariogenic bacteria. It is a correct overview that helps to know the current literature in this particular topic.

Specific Comments:

Line 275. A dot is missing.

Author Response

Point:

Line 275. A dot is missing.

Reply:

Thank you for your comments and suggestions.

The missing dot was added (now in Line 284).

A revised manuscript attached.

Reviewer 3 Report

The review aims to investigate the mechanisms of bioactive glass on the management of dental caries. The topic is very interesting and well discussed. However, a paragraph about the current materials used as fillers for dental caries treatment should be introduced.

In the manuscript there are some mistakes, one of that is Ca+ (correct in Ca2+).

Furthermore, the manuscript lacks in images (i.e. X-ray, Raman or SEM). The authors, should introduce some of that images.

Author Response

Point:

A paragraph about the current materials used as fillers for dental caries treatment should be introduced. In the manuscript there are some mistakes, one of that is Ca+ (correct in Ca2+). Furthermore, the manuscript lacks in images (i.e. X-ray, Raman or SEM). The authors should introduce some of that images.

Reply:

Thank you for your comments and suggestions.

- I went through the whole manuscript again and corrected all the mistakes.

- As suggested, some of the current materials used in dental fillings (page 2, lines 59-63) are added: “For the restorative method metioned above, various materials are used. These include chemically bonded ceramic cements set by acid-base reaction such as zinc phosphate, silicate, polycarboxylate and glass ionomers. Composite resin is another type of cement that set by a polymerization reaction. Besides, resin modified glass-ionomer cement is made by combining the two reactions [7].” There is rearrangement of the paragraphs in pages 1-2 and renumbering of the references.

- In page 12, line 268-270, two SEM images are added: “Figure 2 SEM images of the morphology of demineralized dentine: (a) 10000× magnification image of demineralized dentine treated with bioactive glass; (b) 10000× magnification image of demineralized dentine without treatment with bioactive glass. ”

I cannot provide XRD and Raman images in this review manuscript, because I did not have these images from my own research. These images can be found in the published papers (references 17, 25, 26) which I cited.

The revised manuscript is attached. 

Round 2

Reviewer 3 Report

The revised manuscript is suitable for the publication.